



# The temperature change shortcut: effects of mid-experiment temperature changes on the deformation of polycrystalline ice

Lisa Craw[1], Adam Treverrow[1], Sheng Fan[2], Mark Peternell[3], Sue Cook[4], Felicity McCormack[5], and Jason Roberts[6,4]

[1]Institute for Marine and Antarctic Studies, University of Tasmania, Hobart, TAS, Australia
[2]Department of Geology, University of Otago, Dunedin, New Zealand
[3]Department of Earth Sciences, University of Gothenburg, Gothenburg, Sweden
[4]Australian Antarctic Program Partnership, Institute for Marine and Antarctic Studies, University of Tasmania, Hobart, TAS, Australia
[5]School of Earth, Atmosphere & Environment, Monash University, Melbourne, VIC, Australia
[6]Australian Antarctic Division, Hobart, TAS, Australia

**Correspondence:** Lisa Craw (lisa.craw@utas.edu.au)

**Abstract.** It is vital to understand the mechanical properties of flowing ice to model the dynamics of ice sheets and ice shelves, and to predict their behaviour in the future. We can do this by performing deformation experiments on ice in laboratories, and examining its mechanical and microstructural responses. However, natural conditions in ice sheets and ice shelves extend to low temperatures ($<-5\,^\circ$C), and high octahedral strains ($>0.08$), and emulating these conditions in laboratory experiments can take

an impractically long time. It is possible to accelerate an experiment by running it at a higher temperature in the early stages, and then lowering the temperature to meet the target conditions once the tertiary creep stage is reached. This can reduce total experiment run-time by $>1000$ hours, however it is not known if this could affect the final strain rate or microstructure of the ice and potentially introduce a bias into the data. We deformed polycrystalline ice samples in uniaxial compression at $-2\,^\circ$C before lowering the temperature to either $-7\,^\circ$C or $-10\,^\circ$C, and compared the results to constant temperature experiments. Tertiary

strain rates adjusted to the change in temperature very quickly (within 3% of the total experiment run-time), with no significant deviation from strain rates measured in constant-temperature experiments. In experiments with a smaller temperature step ($-2\,^\circ$C to $-7\,^\circ$C) there is no observable difference in the final microstructure between changing-temperature and constant-temperature experiments which could introduce a bias into experimental results. For experiments with a larger temperature step ($-2\,^\circ$C to $-10\,^\circ$C), there are quantifiable differences in the microstructure. These differences are related to different

recrystallisation mechanisms active at $-10\,^\circ$C, which are not as active when the first stages of the experiment are performed at $-2\,^\circ$C. For studies in which the main aim is obtaining tertiary strain rate data, we propose that a mid-experiment temperature change is a viable method for reducing the time taken to run low stress and low temperature experiments in the laboratory.





# 1 Introduction

## 1.1 Background

Ice is a mechanically anisotropic material, meaning that its mechanical properties change with direction. During deformation, it undergoes microstructural changes in response to changing stress and temperature conditions. This microscale anisotropy leads to large-scale anisotropy of larger ice masses like ice shelves and streams, and affects their response to external changes such as those related to climate change (Castelnau et al., 1998; Harland et al., 2013). This effect is dramatic; strain rates for anisotropic polycrystalline ice can be an order of magnitude higher than those for isotropic ice under the same conditions (Gao 25 and Jacka, 1987; Treverrow et al., 2012). Measuring the mechanical and microstructural properties of deforming ice under different conditions is an important process, but it can take an unreasonable length of time (months to years).

Laboratory deformation experiments allow us to examine the behaviour of ice under specific stress and temperature conditions, and so are an invaluable tool for understanding ice flow on a small scale. The temperature of ice during deformation can significantly affect both strain rate and microstructural characteristics such as crystallographic preferred orientation (CPO). 30 Studying the mechanical and microstructural response of ice to stress under changing temperature conditions will allow us to examine how long the effects of previous temperature conditions persist as deformation proceeds, and evaluate the effect this may have on experimental design. The aims of this study are to establish the extent to which microstructural characteristics of laboratory ice deformed to tertiary creep at one temperature persist once the temperature changes, and to outline a robust way to perform ice deformation experiments more quickly without compromising their results.

## 35 1.2 Ice creep

From experiments on laboratory-made ice, we have a good understanding of how pure ice with an initially isotropic microstructure deforms (e.g. Kamb, 1972; Budd and Jacka, 1989; Montagnat et al., 2015; Peternell and Wilson, 2016; Vaughan, 2016). When a stress is applied to a mass of polycrystalline ice, it will deform viscously in response to that stress. This deformation, or "creep", behaviour changes depending on temperature, strain rate and total strain, as the microstructure of the ice changes. 40 As shown in Fig. 1, when a constant stress is applied to a piece of initially isotropic ice, it experiences three main stages of creep: a decelerating creep rate following the initial elastic deformation (primary creep); a period of constant minimum strain rate (secondary creep); and then an acceleration before a quasi-constant higher strain rate is reached (tertiary creep) (Budd and Jacka, 1989).

These stages of creep are associated with distinct stages of microstructural behaviour in the ice:

– *Primary creep*: incompatible dislocations in the crystal lattice intersect to form "dislocation tangles", preventing further lattice deformation. The strain rate decreases as the rate of deformation is controlled by crystals which are unfavourably oriented for creep (Duval et al., 1983). There is no other significant change in microstructural characteristics during this creep stage (Gao and Jacka, 1987; Vaughan, 2016).

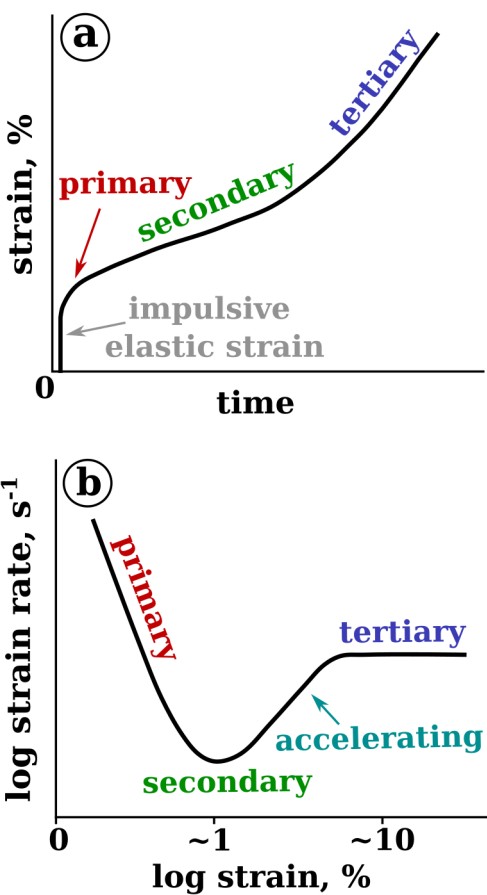

**Figure 1.** Plots adapted from Budd and Jacka (1989) and Durham et al. (2010) showing the primary, secondary and tertiary stages of creep with respect to strain over time (a) and strain rate over strain (b) in a constant load experiment.

- – *Secondary creep*: new grains begin to nucleate, which are free of dislocation tangles and are favourably oriented for

deformation under the imposed conditions. A minimum strain rate is reached as this process is balanced with strain hardening (Wilson et al., 2014).

- – *Acceleration and tertiary creep*: A crystallographic preferred orientation (CPO) begins to form, causing softening and strain rate increase. A quasi-constant strain rate is then established, which does not change significantly beyond this point as long as the conditions of deformation remain the same. Eventually a "steady-state" microstructure is reached,

where continuous dynamic recrystallisation and strain hardening are balanced and grain sizes are constant (Wilson et al., 2014), however this can be delayed significantly beyond the establishment of a quasi-constant strain rate. While tertiary creep is generally reached at octahedral strains of $5 - 10\%$, in some experiments deformed to very high strains (>57%





shortening) a steady-state microstructure has not been firmly established by the end of the experiment (Peternell et al., 2019).

Strain rates at the primary, secondary and tertiary creep stages are controlled by the specific stress and temperature conditions, as well as microstructural and chemical characteristics of the ice (Gao and Jacka, 1987; Treverrow et al., 2012; Hammonds and Baker, 2018). Because ice in many natural scenarios has been flowing for some time and typically has already reached a quasi-constant tertiary strain rate (except in some key regions where assumptions of tertiary creep are not valid (Budd et al., 2013; Graham et al., 2018)), the accelerating and tertiary creep stages are of interest to many glaciologists (Gao

and Jacka, 1987). However, the versions of the Glen flow relation most commonly used in ice dynamics modelling (Glen, 1952, 1955; Nye, 1953) are derived from the secondary creep stage, which takes less time to reach in an experiment. A method to more easily measure tertiary creep rates would be very useful for parameterising a flow law based on tertiary creep rates.

In particular, knowledge of secondary and tertiary strain rates is useful for determining enhancement factors for flow laws used in ice sheet models (e.g. Greve and Blatter, 2009). Furthermore, understanding how these enhancement factors vary depending on the underlying stress configurations gives scope for understanding key features of ice deformation, including anisotropy (Budd et al., 2013; Graham et al., 2018).

### 1.3   Microstructural development

It has been repeatedly observed in ice (e.g. Piazolo et al., 2013; Montagnat et al., 2015; Qi et al., 2017) and rocks (e.g. Avé Lallemant, 1985; Stipp et al., 2002; Little et al., 2015) that the stress and temperature conditions present during deformation affect the microstructure of the material. In experimentally deformed ice and quartz aggregates and natural quartz veins, bulging (BLG) and subgrain rotation (SGR) recrystallisation are the dominant recrystallisation mechanisms at relatively lower temperatures. As temperature and stress increases, grain boundary migration (GBM) becomes more dominant (Hirth and Tullis,

1992; Stipp et al., 2002). Most experiments in ice, for practical reasons, are performed at high homologous temperatures (typically $>-10\,°C$) and low stresses, where grain boundary migration is dominant. This produces a characteristic microstructure of hollow cone CPOs, and irregularly shaped grains with interlocking boundaries (see e.g., Wilson et al., 2014; Montagnat et al., 2015). When the temperature is decreased, the strength of the CPO decreases in experiments performed at temperatures approaching $-15\,°C$, as BLG and SGR become more dominant mechanisms, and CPOs tend toward clusters rather than cones

(Jacka and Jun, 2000; Qi et al., 2017). Active GBM allows microstructures to change rapidly, within strains of $\sim 0.1$, while lattice rotation appears to be a slower-acting recrystallisation mechanism (De La Chapelle et al., 1998; Montagnat et al., 2015). It is important to understand the changing characteristics of ice microstructure at a wide range of temperatures and differential stresses, and at all stages of creep, as it can have a significant effect on the rheological behaviour of the ice (Piazolo et al., 2013). When evaluating the effects of changing temperature on rheology, we must consider any lasting effects on the microstructure

of the ice.





### 1.4 Temperature changes and experimental design

Laboratory deformation experiments provide an opportunity to replicate natural conditions of ice deformation, varying conditions such as temperature and stress to examine their effects on flow behavior. However, running an experiment through to the tertiary creep stage at strain rates approaching those found in most natural scenarios can take an impractically long time.
Consequently, the strain rates and other measurements from experimental studies are often extrapolated to compare them with in situ data in a way that may not be robust. This study is designed to address that problem by assessing whether it is possible to reduce the time taken to complete an experiment by running it at a higher temperature during primary and secondary creep, and then lowering the temperature to emulate the target conditions once the tertiary creep stage has been reached.

It has been demonstrated in both natural ice (Russell-Head and Budd, 1979; Gao and Jacka, 1987) and laboratory ice
(Treverrow et al., 2012) that once ice has been deformed through to tertiary strain, if it is deformed again under similar conditions it will progress straight from the initial elastic deformation stage to resume deformation at the same constant tertiary strain rate, with no significant change in CPO, allowing tertiary creep to be reached within strains of $2 - 3\%$. However, if the stress configuration is changed in the second stage of the experiment, characteristics of the original CPO can persist to higher strains (Budd and Jacka, 1989).

Russell-Head and Budd (1979) reduced the time required to obtain secondary minimum strain rates in a series of shear experiments by increasing and decreasing temperatures within experiments. They achieved this by beginning each experiment at higher temperatures of $-2$ or $-5\,^{\circ}\mathrm{C}$, running it through to the secondary minumum (strains of $\sim 0.01$), and then stepping the temperature down to $-5, -10, -15$ and $-20\,^{\circ}\mathrm{C}$, accumulating shear strains of at least $0.001$ at each step. This allowed them to gather minimum strain rate data at a range of temperatures within a single experiment. Treverrow (2009) used a similar method
for a series of horizontal shear experiments, stepping through $-2, -5, -10, -15$ and $-20\,^{\circ}\mathrm{C}$ to gather strain rate information at each step. Total accumulated strains were kept to $0.02 - 0.03$ to minimise any microstructural evolution, as deformation did not progress far beyond the secondary minimum.

So far, there has not been a systematic study undertaken on the effects of changing the temperature once the tertiary creep stage has been reached. The purpose of this study is to compare the microstructural and mechanical data from compression
experiments conducted at a single temperature and experiments conducted at multiple temperatures, and evaluate whether this method compromises the results by introducing any systematic bias into the strain rate and microstructure data.

## 2 Methods

### 2.1 Laboratory

The samples used in this study were initially isotropic polycrystalline pure water ice, prepared using the methods described by
Jacka (1984) and Treverrow et al. (2012). Pure de-ionised water was frozen into blocks and passed through an industrial food processor to produce seed grains, which were then sieved to separate out size fractions of $250 - 425\,\mu\mathrm{m}$ and $425 - 1800\,\mu\mathrm{m}$. These two size fractions were combined in equivalent volumes. The seed grains were poured into a mold, which was then





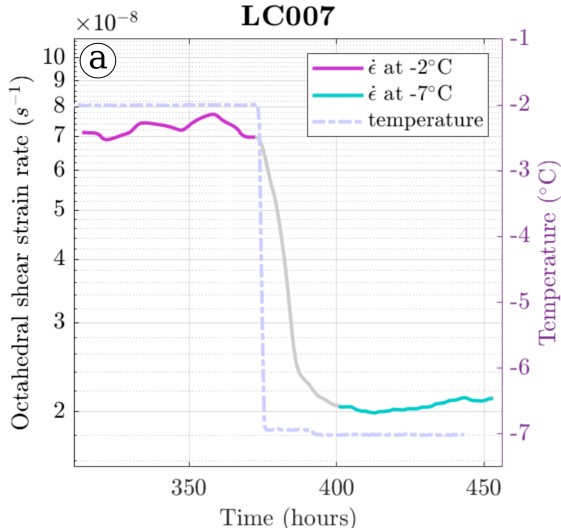

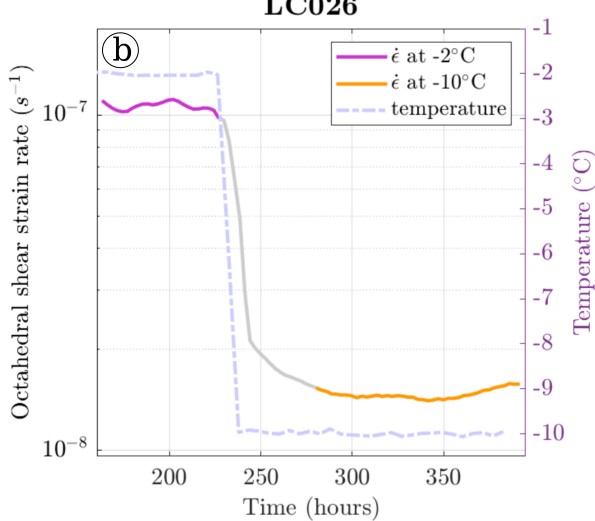

**Figure 2.** Plots of octahedral shear strain rate *vs.* time for (a) **LC007** ($-2\,°\mathrm{C}$ to $-7\,°\mathrm{C}$), and (b) **LC026** ($-2\,°\mathrm{C}$ to $-10\,°\mathrm{C}$).



| | $T$, °C | $t$, hours | $\epsilon_{max}$ | $\dot{\epsilon}_{sec}, s^{-1}$ | $\dot{\epsilon}_{tert1}, s^{-1}$ | $\dot{\epsilon}_{tert2}, s^{-1}$ | $gs_{med}, \mu m$ | $IQR, \mu m$ | $J\text{-}index$ |
|---|---|---|---|---|---|---|---|---|---|
| **LC001** | -2 | 498 | 0.1333 | $3.21\times10^{-8}$ | $9.36\times10^{-8}$ | - | 941 | 1341 | 3.57 |
| **LC002** | -2 | 613 | 0.14164 | $2.89\times10^{-8}$ | $7.78\times10^{-8}$ | - | 993 | 1802 | 4.99 |
| **LC004** | -7 | 2080 | 0.1633 | $8.88\times10^{-9}$ | $3.00\times10^{-8}$ | - | 682 | 610.9 | 2.83 |
| **LC005** | -7 | 1913 | 0.1385 | $8.94\times10^{-9}$ | $2.64\times10^{-8}$ | - | 906 | 1183 | 3.38 |
| **LC006** | -2, -7 | 899 | 0.1377 | $3.27\times10^{-8}$ | $7.73\times10^{-8}$ | $2.13\times10^{-8}$ | 984 | 1299 | 3.55 |
| **LC007** | -2, -7 | 881 | 0.1305 | $2.77\times10^{-8}$ | $8.24\times10^{-8}$ | $2.13\times10^{-8}$ | 1200 | 1462 | 3.95 |
| **LC009** | -2 | 266 | 0.0860 | $3.00\times10^{-8}$ | $9.04\times10^{-8}$ | - | 1070 | 1429 | 3.49 |
| **LC021** | -10 | 1442 | 0.0570 | $6.18\times10^{-9}$ | $1.77\times10^{-8}$ | - | 867 | 948.8 | 1.98 |
| **LC023** | -10 | 1599 | 0.0736 | $6.76\times10^{-9}$ | $2.07\times10^{-8}$ | - | 561 | 420.7 | 1.95 |
| **LC025** | -2, -10 | 687 | 0.0999 | $4.27\times10^{-8}$ | $1.02\times10^{-7}$ | $1.38\times10^{-8}$ | 929 | 1144 | 3.61 |
| **LC026** | -2, -10 | 688 | 0.0947 | $3.56\times10^{-8}$ | $1.08\times10^{-7}$ | $1.53\times10^{-8}$ | 805 | 913.1 | 4.06 |

**Table 1.** List of all experiments performed, along with their starting parameters, measured strain rates and microstructural properties. $t$ is the total experiment run-time excluding set-up and decommission, $\epsilon_{max}$ is total accumulated strain, $\dot{\epsilon}_{sec}$ is the secondary strain rate, $\dot{\epsilon}_{tert1}$ and $\dot{\epsilon}_{tert2}$ are tertiary strain rates measured at the first and second (if applicable) temperatures respectively, $gs_{med}$ and $IQR$ are the median value and interquartile range of the measured grain sizes after deformation, and $J\text{-}index$ is a measure of $c$-axis orientation density as decribed by Bunge (1983).

flooded with water at $0\,°C$, and carefully agitated to remove bubbles. The insulated mold was then left in a $-3\,°C$ freezer for several days to freeze.

Samples were cut using a bandsaw into rectangular blocks with approximate dimensions of $45\times90\times50\,mm$, lightly sanded to remove marks from the bandsaw blade, and frozen into aluminium platens at the top and bottom to leave $\sim50\,mm$ sample height exposed for deformation. The samples were installed into deformation rigs described by Jun et al. (1996) and Treverrow et al. (2012), and deformed through uniaxial compression by loading with lead weights from above. To approximate a near-constant octahedral stress of $0.25\,MPa$, assuming a constant rate of increase in cross-sectional area and conservation of volume,
loads were increased periodically (every 2-5 days).

Vertical displacement was logged at a frequency of $0.05\,Hz$ using digital dial indicators, and the sample was kept at a constant temperature in a bath of circulating 1-1.5 cSt viscosity silicone oil within a chest freezer, heated by thermistor-controlled elements. Data from the experiments were periodically retrieved and analysed during the course of the experiments. At the conclusion of each experiment, the temperature in the bath was lowered to $<-18°\,C$ over a timespan of four to six hours,
and the samples removed from the rigs within the following week.

After extraction from the deformation rigs, samples were cut using a bandsaw to expose a vertical face containing the axis of compression. This face was sanded to remove marks from the bandsaw blade, and then thermally bonded onto $10\,cm^2$ glass slides. Excess ice was removed above the surface of the slide using a motorised microtome, to produce thin sections $\sim500\mu m$





thick. These sections were loaded into a custom-built Russell-Head Instruments section viewer and G50 fabric analyser (Wilson
et al., 2003, 2007), and scans of the deformed sections collected at a resolution of $20\,\mu\mathrm{m}$.

A complete list of experiments and their parameters is shown in Table 1. A series of control experiments were carried out at
constant temperatures of $-2^\circ\mathrm{C}$ (LC001 and LC002 and LC009), $-7^\circ\mathrm{C}$ (LC004 and LC005) and $-10^\circ\mathrm{C}$ (LC021 and LC023)
until a constant tertiary strain rate was well established, and microstructural data collected. Four experiments were run through
into tertiary strain at $-2^\circ\mathrm{C}$, and then the temperature was lowered to either $-7^\circ\mathrm{C}$ (LC006 and LC007) or $-10^\circ\mathrm{C}$ (LC025 and
LC026) and left to run until a new stable tertiary strain rate had been established (typically a further 0.03-0.04 accumulated
strain). After the temperature was changed, a tertiary strain rate reflective of the new temperature was reached within a span of
30 hours for the $-2^\circ\mathrm{C}$ to $-7^\circ\mathrm{C}$ experiments, and 60 hours for the $-2^\circ\mathrm{C}$ to $-10^\circ\mathrm{C}$ experiments (see Fig. 2). This represents
less than 3% of the total run-time of the changing-temperature experiments. For Set 1, the changing-temperature experiments
ran for an average of 890 hours compared with the constant temperature $-7^\circ\mathrm{C}$ experiments which ran for an average of
1997 hours. For Set 2, the changing-temperature experiments ran for an average of 688 hours, compared with the constant
temperature $-10^\circ\mathrm{C}$ experiments which ran for an average of 1521 hours. For both sets, the changing-temperature experiments
had a shorter run-time by 55%.

## 2.2 Mechanical data processing

Erroneous displacement and temperature values were removed by filtering for consecutive values with a difference greater
than a minimum step size (an appropriate mimimum step size was selected for each experiment based on visual inspection).
Sudden jumps in displacement from load increases and disturbances to the apparatus were removed manually. Strains were
calculated using the difference between consecutive displacement data points, and strain rates were calculated incrementally
over intervals of between 200 and 2000 displacement and time data points (greater intervals were used at lower strain rates).
Octahedral shear strains ($\tau_{oct}$) were derived from the applied compressive stresses ($\sigma'$) using Equation 1 (Nye, 1953):

$$\tau_{oct} = \frac{1}{\sqrt{3}}\sigma' \tag{1}$$

Smoothed strain rates were then derived using a cubic spline fit, with a smoothing parameter of $0.01$. Octahedral shear strain
rates were extracted from the smoothed data for the secondary and tertiary creep stages by averaging values within ranges
manually selected from visual inspection of the plotted data. Experiments which reached higer total accumulated strains (>0.08
at a single temperature) tend to show a drop-off in tertiary strain rate, as samples expand unevenly and the assumption of a
constant rate of increase in cross-sectional area becomes less appropriate. In these cases, the tertiary strain rate value was taken
from data points closer to the beginning of tertiary creep, before the drop in values.

## 2.3 Microstructural data processing

The procedures used for processing the microstructural data are novel, and are described in detail in Appendix A1. In brief,
the raw orientation data from the G50 fabric analyser were first converted to a data format readable by MATLAB®. The data
were then filtered to remove anomalies at grain boundaries and other areas of low data quality, and the remnant grains were





reconstructed to fill the sample area, using methods based on the FAME (Fabric Analyser Based Microstructure Evaluation) program (Hammes and Peternell, 2016) and functions of the MTEX toolbox (Bachmann et al., 2010; Mainprice et al., 2015). We extracted grain size (equal area diameter), shape-preferred orientation (SPO), and $c$-axis CPO (one point per grain) from the processed fabric analyser data. The $c$-axis CPO was contoured from the $c$-axis pole figure with a kernel half-width of $7.5°$.

We quantified the intensity of the $c$-axis CPO using the PfJ-index (Kilian and Heilbronner, 2017).

## 3  Results

### 3.1  Mechanical data

Plots of octahedral strain rate data for all experiments are shown in Fig. 3. Experiments have been divided into Set 1 (for comparison of $-2°C$ to $-7°C$ temperature changes), and Set 2 (for comparison of $-2°C$ to $-10°C$ temperature changes). When

comparing the strain rates and final microstructure of changing-temperature experiments to constant-temperature experiments, we will only compare samples which reached a similar total strain, to avoid considering the influence of accumulated strain on behaviour. Therefore, the $-2°C$ control samples for the first set of experiments are **LC001** ($\epsilon_{max} = 0.133$) and **LC002** ($\epsilon_{max} = 0.142$), and for the second set only **LC009** ($\epsilon_{max} = 0.086$) will be used for comparison, as experiments running at $-10°C$ cannot be run to as high total strains within a reasonable timeframe.

There is good agreement between duplicate experiments, with strain rates from different experiments differing on average by $±10\%$ when running at the same temperature. The level of variation in tertiary strain rates (Table 1) between changing-temperature experiments and constant-temperature experiments is within the level of variation between duplicate experiments.

### 3.2  Microstructural data

Microstructural data collected from all samples are shown in Fig. 4 (Set 1) and Fig. 5 (Set 2). Median values and interquartile

ranges of grain size distributions, alongside J-indices for $c$-axis CPOs are also listed in Table 1. The starting material, known as 'standard laboratory ice' (Fig. 4, top left), is made up of polygonal grains with straight boundaries, and has no crystallographic preferred orientation (J-index = 1.18). Grains have a mean size of $1460\,\mu m$, with an interquartile range of $1087\,\mu m$. With this starting point as a reference, we will consider the microstructural data from both sets of experiments in turn.

**Set 1** ($-2°C$ **to** $-7°C$)**:**

All deformed samples in this set are composed of irregularly-shaped grains with interlocking boundaries. All samples after deformation have a strong cone CPO (J-indices $2.83 - 4.99$) with the majority of $c$-axes oriented $10 - 30°$ from the compression direction. Median grain sizes for all samples lie within a range of $680 - 1200\,\mu m$, with interquartile ranges $610 - 1801\,\mu m$. The J-indices, median grain sizes and grain size interquartile ranges of the two changing-temperature experiments lie within the ranges of the same values for the constant-temperature experiments in this set.

**Set 2** ($-2°C$ **to** $-10°C$)**:**



**Figure 3.** Smoothed octahedral shear strain rate data plotted against total accumulated octahedral shear strain for all experiments. (a), (c): constant-temperature experiments, (b), (d): changing-temperature experiments.




**Figure 4.** Results from microstructural analysis of all samples in Set 1 (−2 °C to −7 °C), alongside a sample of undeformed standard ice. Shown for each sample, from left to right: sample number and J-index above a lower-hemisphere stereonet plot of *c*-axis orientations of 5000 randomly selected pixels, alongside a contoured plot of the same data (scale shown in the upper right); thin section image from processed fabric analyser data, coloured by *c*-axis orientations according to the legend (upper right).


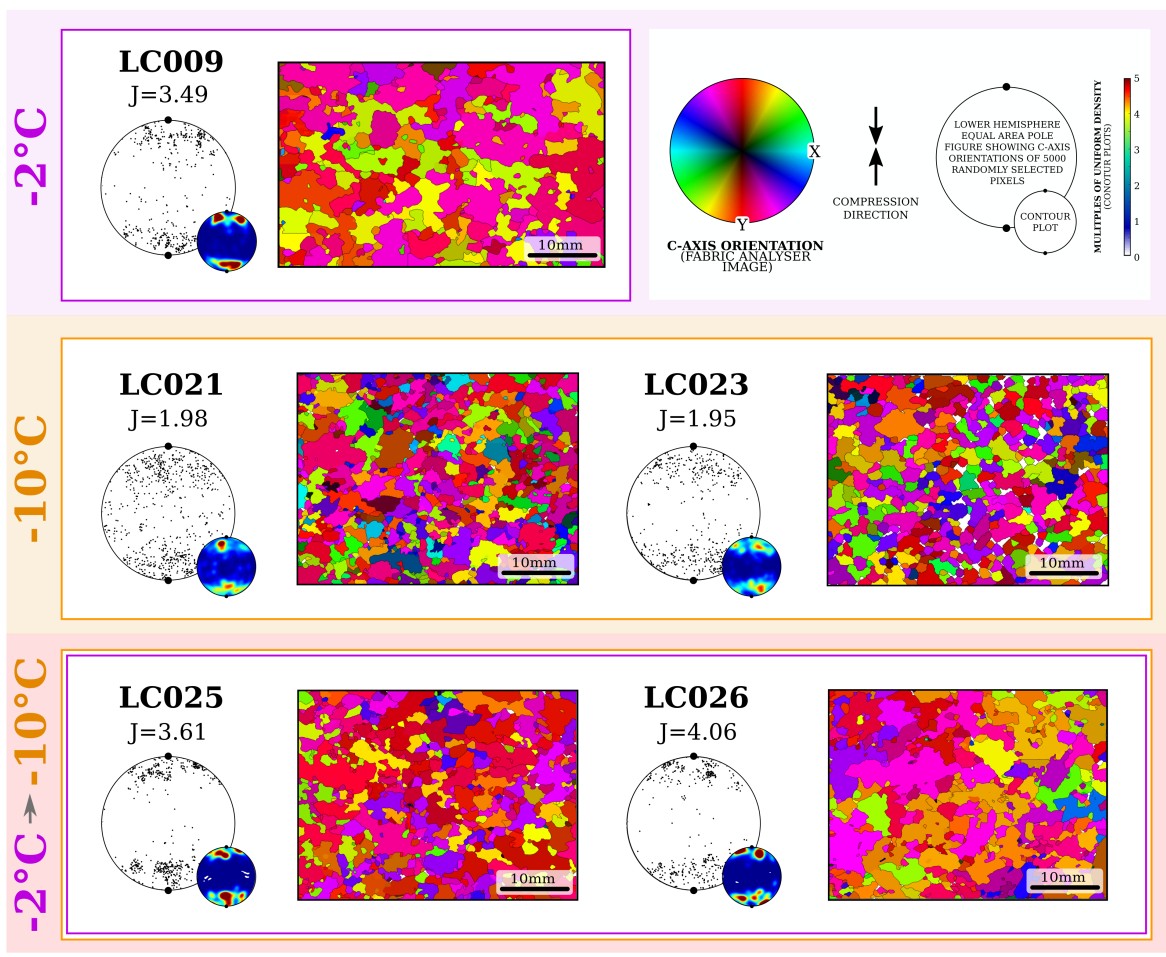

**Figure 5.** Results from microstructural analysis of all samples in Set 2 ($-2\,^\circ$C to $-10\,^\circ$C). Shown for each sample, from left to right: sample number and J-index above a lower-hemisphere stereonet plot of $c$-axis orientations of 5000 randomly selected pixels, alongside a contoured plot of the same data (scale shown in the upper right); thin section image from processed fabric analyser data, coloured by $c$-axis orientations according to the legend (upper right).





The constant-temperature $-2\,^{\circ}$C sample and the changing-temperature samples for this set have the same microstructural characteristics as those from the previous set. All have irregularly-shaped grains with interlocking boundaries and a strong $10-30\,^{\circ}$ cone CPO centred around the compression direction (J-indices $3.49-4.05$), median grain sizes in the range $805-1070\,\mu$m, and grain size interquartile ranges of $913-1340\,\mu$m. However, the constant-temperature $-10\,^{\circ}$C samples have a
significantly different microstructure. Most strikingly, their $c$-axis CPOs, while still vertical cones, are approaching clusters and are much weaker than the other samples in this set, with J-indices of $1.95$ and $1.98$. Their median grain sizes ($561$ and $867\,\mu$m) and interquartile ranges ($949\,\mu$m and $421\,\mu$m) overlap with the range of values measured in other samples, but are at the smaller extremity.

All samples measured from both sets show a similar grain size distribution, with a unimodal distribution skewed toward
smaller sizes and a decreasing "tail" extending to larger sizes.

## 4 Discussion

A comparison of strain rates at the two temperatures suggests that there is no significant effect of previous temperature history on tertiary strain rate. Strain rate values for experiments performed under identical conditions lie on average within $\pm10\%$, where any larger variations are explained by uncertainties and assumptions made during the experimental process
(*e.g.*, constraints on temperature control, uncertainties in sample measurement, and the assumption that displacement is evenly distributed down the length of the sample). Tertiary strain rates at both $-7\,^{\circ}$C and $-10\,^{\circ}$C from the changing-temperature experiments agree with those from their equivalent constant-temperature experiments to within the same level of variability, meaning that it is not possible to detect any effect of the previous temperature history of the samples on tertiary strain rate using these experimental methods. Once the temperature has been changed, it is a matter of tens of hours (out of a several hundred- to
thousand-hour experiment) before deformation continues at a quasi-constant strain rate with no obvious perturbations or delay in strain rate response.

The microstructural characteristics observed in these samples after deformation are comparable to those from other compression experiments in the literature. The development of a vertical small-circle girdle CPO centred around the compression direction has been observed many times in polycrystalline ice above $-15\,^{\circ}$C (*e.g.* Kamb, 1972; Jacka, 1984; Treverrow et al.,
2012; Wilson et al., 2014), and the interlocking, irregular grain boundaries seen in all deformed samples in this study are comparable to those observed by Montagnat et al. (2015) and Vaughan (2016) after similar experiments. Mean grain sizes fall within a range consistent with those observed by Jacka and Jun (1994) for tertiary steady-state crystal sizes. The distributions of grain sizes in deformed samples are consistent with those recorded by Stipp et al. (2010) for polycrystalline materials which have undergone dynamic recrystallisation dominated by grain boundary migration (GBM) mechanisms, as expected in high-temperature ice (Qi et al., 2017).

It is not possible to distinguish between changing-temperature and constant-temperature experiments in Set 1 on the basis of microstructure. As the stress conditions (unconfined vertical compression at $0.25\,$MPa) are the same, and the temperatures ($-2\,^{\circ}$C and $-7\,^{\circ}$C) are very close, the microstructure that develops during deformation is too similar to be distinguished





using these methods. Mean steady-state grain sizes in polycrystalline materials undergoing dynamic recrystallisation have been observed to adjust according to experimental temperature (Jacka and Jun, 1994; Cross et al., 2017; Treverrow, 2009; Stipp et al.,

2010), however the level of variability in grain size data across all samples means that it is not possible to distinguish between samples deformed at these temperatures. This means that there is no obvious disadvantage to performing these experiments at a higher temperature initially, saving over 1000 hours (55%) of experiment run-time to derive a reliable tertiary strain rate at $-7\,^{\circ}$C.

In the samples from Set 2 there are clear microstructural differences between constant-temperature and changing-temperature

experiments. In samples where the temperature was dropped from $-2\,^{\circ}$C to $-10\,^{\circ}$C after the onset of tertiary creep, the microstructure is still much more comparable to experiments conducted completely at $-2\,^{\circ}$C, failing to match those conducted entirely at $-10\,^{\circ}$C despite strain rates rapidly adjusting to match the new conditions. Under deformation with no confining pressure at temperatures greater than approximately $-15\,^{\circ}$C, ice generally develops a microstructure characteristic of active GBM and BLG. However, in some lower-temperature experiments the strength of the CPO decreases with decreasing temper-

ature. This is due to the increasing contribution of lattice rotation and polyganisation as recrystallisation mechanisms, even while GBM is still present and even dominant. Our data suggest that the characteristics of the microstructure which develops during all three creep stages at $-2\,^{\circ}$C, at which temperature GBM is overwhelmingly dominant as a recrystallisation mechanism, are sufficiently different from those present after constant temperature experiments at $-10\,^{\circ}$C. Those characteristics persist once the temperature has been lowered and some further strain ($0.02 - 0.03$) has been accumulated. It is possible that if

a larger amount of strain were accumulated during tertiary creep at $-10\,^{\circ}$C, the microstructure would further adjust to the new conditions, however this would take longer and therefore reduce the usefulness of this method as a way to decrease experiment run time.

## 5   Conclusions

Strain rate data from compression experiments on standard laboratory ice show that tertiary strain rates adjust very quickly

(within 3% of total experiment run time) to a change in temperature, with no obvious lasting effects resulting from the temperature history of the sample. Therefore, we suggest that for experiments where strain rate data are the main object, a mid-experiment temperature change is a viable way to decrease experiment duration with a temperature change step up to and possibly exceeding $8\,^{\circ}$C in magnitude. This method is best used when the main objective of an experiment is to measure tertiary strain rate data under different experimental conditions, for example when deriving flow law enhancement factors for use

in ice shelf and ice sheet models. In this context it can save 55% of experiment run time, allowing data to be collected under a far wider range of conditions than has previously been practical.

Differences in final sample microstructure between changing-temperature and constant-temperature experiments are not detectable in high temperature experiments with a small temperature step ($-2\,^{\circ}$C to $-7\,^{\circ}$C), therefore there is no observable disadvantage to using the temperature-change method at these temperatures. However, microstructural data where the change

in temperature is $8\,^{\circ}$C in magnitude (in this case changing from $-2\,^{\circ}$C to $-10\,^{\circ}$C) may not be truly representative of their





final experimental temperature. Even when changing between temperatures where the same recrystallisation mechanisms are dominant, the relative contribution of mechanisms can change significantly, resulting in quantifiably different microstructural characteristics.

These results also show that the tertiary strain rate of deforming ice will adjust almost instantaneously to a change in temperature, a fact which should hold true in natural scenarios. Regardless of the microstructure, using this method will allow strain rate data at much more realistic strain rates and stresses to be derived at low temperatures on a laboratory timescale, allowing the paramaterisation of ever more accurate ice flow laws.

*Data availability.* Data can be obtained at doi:10.4225/15/58eedf0d72be9 (Craw et al., 2020).

## Appendix A: Microstructural data processing

The workflow of microstructural data processing is shown in Fig. A1. The process uses MTEX, but is closely modelled on FAME (Fabric Analyser Based Microstructure Evaluation), a good alternative which is compatible with G60 data. MTEX (Bachmann et al., 2010; Mainprice et al., 2015) is a MATLAB®-based toolbox which has been widely used to analyse ice crystallographic textures measured using Electron Backscatter Diffraction (EBSD) (e.g. Prior et al., 2015; Qi et al., 2017, 2019). FAME is a comprehensive MATLAB®-based software which has been widely used to quantify thin section data from
ice (e.g. Peternell et al., 2014; Hammes and Peternell, 2016), and minerals such as quartz (Peternell et al., 2010; Rodrigues et al., 2016; Zibra et al., 2017) and calcite (Köpping et al., 2019). In this study, we used a FAME-based method to to convert the raw binary data from the G50 fabric analyser (.cis file format) to text files which could then be imported into the MTEX toolbox in MATLAB®. The converted raw data were then processed and statistically analysed using scripts developed using the MTEX toolbox and FAME program for reference.

The raw output from a G50 fabric analyser is a binary .cis file containing the positions in x-y co-ordinates, $c$-axis orientations in Clar-notation (dip direction and dip), geometric quality, retardation quality, and other parameters for each pixel in the field of view (Peternell et al., 2014). Geometric quality and retardation quality quantify the data quality of each $c$-axis orientation measured with a range of 0 (bad) to 100 (excellent) (Peternell et al., 2009). We used FAME to convert the $c$-axis orientations from Clar-notation to Euler angles (phi1, Phi, phi2), with phi2 set to 0. The output is an MTEX import text file containing the
.cis binary information and Euler angles (Peternell et al., 2014; Hammes and Peternell, 2016).

The converted raw data were filtered using scripts developed using the MTEX toolbox in MATLAB®. We applied the "grain reconstruction" function (Bachmann et al., 2011) to pixels with geometric quality of $\geq 1$. Grain boundaries were defined where misorientations of neighbouring pixels were larger than $3\,°$. Grain and sub-grain boundaries, bubbles and kink bands can have a non-negligible impact on the quality of reconstructed grain data, introducing artificial fine and elongated grains at boundaries.
The raw data filtering removes very fine grains ($< 30\,\mu$m), and elongated small grains (those with an aspect ratio $< 3$ and equivalent radius $< 100\,\mu$m).





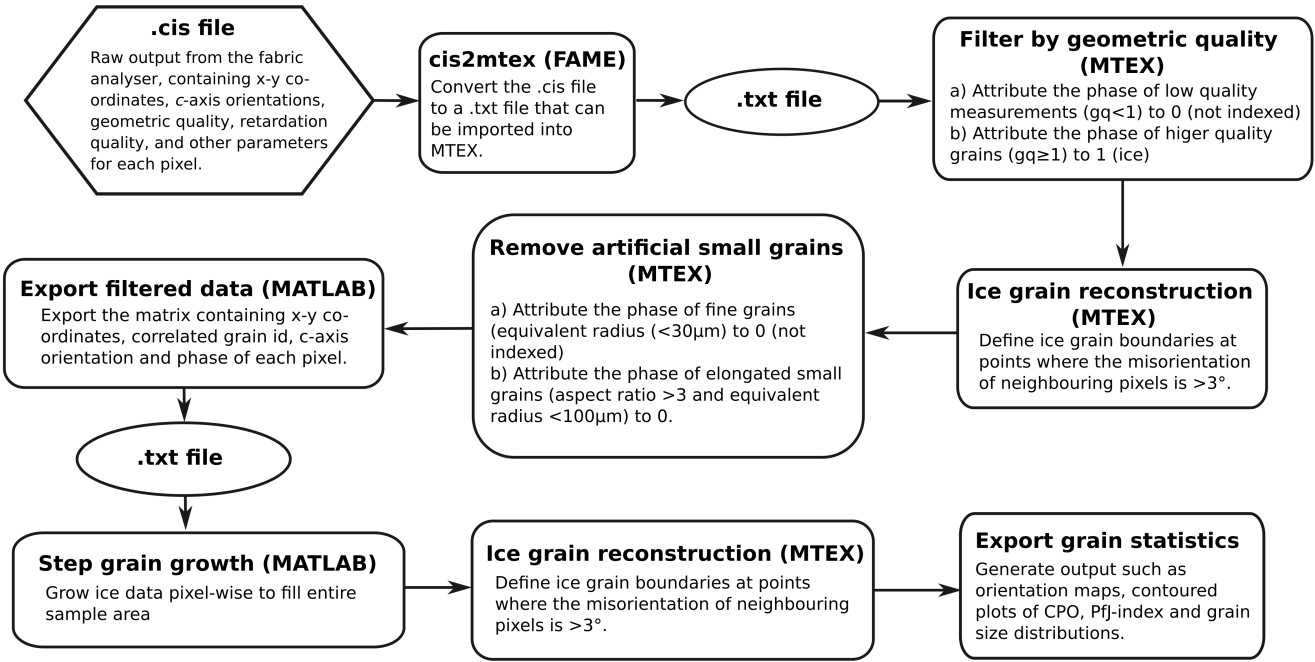

**Figure A1.** Workflow for microstructural data processing

The raw data filtering leaves blank areas along grain boundaries and within grains where artefacts have been removed. We applied a step grain growth function (Hammes and Peternell, 2016) to the filtered raw data. Step grain growth allows a good restoration of the ice grain geometry, improving the grain statistics when compared with the original data. Grains were grown

to fill the entire sample area.

*Author contributions.* LC, AT, SC, FSM and JR designed research. LC and AT performed experiments. SF and MP developed methods for microstructural data analysis, and AT designed methods for mechanical data analysis. LC performed data analysis, and wrote the draft. All authors edited the paper.

*Competing interests.* The authors declare that they have no conflict of interest.

*Acknowledgements.* This work was supported under the Australian Research Council's Special Research Initiative for Antarctic Gateway Partnership (Project ID SR140300001), and the Australian Government's Co-operative Research Centres Programme through the Antarctic Climate and Ecosystems Co-operative Research Centre (ACE CRC). Lisa Craw is supported by an Australian Government Research Training



Program Scholarship at the University of Tasmania. During this publication, Sue Cook received support from the Australian Government as part of the Antarctic Science Collaboration Initiative program. Sheng Fan was supported by a University of Otago doctoral scholarship, an

Antarctica New Zealand doctoral scholarship and University of Otago PERT (Polar Environment Research Theme) seed funding.



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
