# Peer review of "The temperature change shortcut: effects of mid-experiment temperature changes on the deformation of polycrystalline ice"

_The Cryosphere, 2020_

## Referee Comment (RC1) · Nicolas Stoll (Referee) · 18 Dec 2020

The manuscript is well written, has clear objectives, and is of interest for the cryo-community. It shows progress in the field of deformation experiments on polycrystalline ice and explores the approach to reduce run-time by changing the temperature at a certain stage. The manuscript is presented in a logical structure and visualizations are well chosen.

Points which could be improved are a clearer use of terminology (microstructure vs. fabric and texture) and a stronger emphasis on the fact that the conditions of the experiments differ from polar ice sheets and are not easily transferable. Furthermore, I

suggest to use SPO data to quantify changes in the texture and to add (more) references to some parts of the discussion.

Furthermore, I suggest some technical corrections. All in all, the manuscript is of high quality and I recommend the publication of this manuscript in TC subject to minor revisions.

Specific comments:

1. Abstract p1. L.2 We can do this by... Is "this" referring to understanding the mechanical properties of flowing ice, modelling of the dynamics of ice sheets, or predicting the behaviour in the future? Please rephrase to enhance clarity.

2. Abstract p.1 l.4 conditions in ice sheets and ice shelves extend to low temperatures (<-5°C). The temperatures in the majority of ice sheets and ice shelves is well below -5°C, a lower temperature value might be more appropriate. See more detailed comment below.

3. Aims of the study are explained on p. 2. l. 32fff and on p.5 l.43ff. They differ in details (e.g., tertiary creep only mentioned on p. 2. l. 32fff), thus it might be appropriate to combine both sections into one, placed at the end of the introduction to increase readability.

4. P.2 l. 45 Primary creep: The definition is a little bit too simplified, maybe include some information from e.g., Faria et al. (2014b): "During the first creep stage, usually called transient or primary creep, the strain rate decreases rapidly. This deceleration is due to work hardening mainly produced by the load transfer from easyglide to hardglide systems and the increasing strain incompatibilities between the grains, which build up internal stresses and localized heterogeneous strains (Wilson, 1986; Petrenko and Whitworth, 1999; Schulson and Duval, 2009; cf. Sect. 2.2), both clearly identified by the formation of the first dislocation walls and subgrain boundaries (Hamann et al., 2007; Sect. 4.1). Primary creep in ice extends to about 1% of strain, irrespective

of temperature or stress (Budd and Jacka, 1989), and a considerable fraction of it consists of a recoverable "delayed-elastic" strain (sometimes also called "anelastic" strain), implying that part of the deformation is recovered after the load is removed, in a relaxation process that can take several hours (Duval, 1978). Budd and Jacka (1989) report primary recoverable strains of 0.15% and 0.30% for isotropic polycrystalline ice at 10 C compressed at 0.2 MPa and 1.0 MPa octahedral stress, respectively. It is believed that the delayed elasticity of ice is mainly caused by the relaxation of internal stresses by dislocation back-gliding (Glen, 1975; Cole, 2004; Schulson and Duval, 2009)."

5. P4. L10fff: The impact of GBM is strong on texture (grain size, grain shape, SPO) and on grain growth and reduction (dynamic grain growth, see Steinbach et al., 2017), but not so much on CPO (fabric)(e.g., Llorens et al., 2016a, b). Terminology is not always used in the same way in earth and material sciences, so please define "microstructure" in the beginning. Otherwise it is difficult to distinguish between microstructure (glaciology: fabric + texture), fabric and texture (see also comment 8).

6. P. 5 l. 25ff: references for examples of other experiments, in-situ data, extrapolations missing.

7. P. 5 l. 43 it should be clarified that the systematic study is undertaken on laboratory ice.

8. It is mentioned on p. 9 l. 101 that SPO data was derived, but this data is not further used in the manuscript. Showing and discussing SPO data would be a good way to further quantify the microstructural changes, i.e. changes in the ice texture. In this case it is possible to visually analyse the microstructure of the thin section images, however, I would suggest to add SPO data to fully fulfil the statement of "quantifiable differences in the microstructure" as described in the abstract on p.1 l. 14. Otherwise, if only CPO-data is used to quantify changes I suggest to rephrase the wording to e.g., "quantifiable differences in the fabric" to avoid confusion.

9. For practical reasons the temperatures of the conducted experiments are rather high compared to temperatures in deep ice sheets (- 30°C - -20°C) (e.g., Dahl-Jensen et al., 1998, Mony et al. (2020) and, to a lesser degree, in ice shelves (-25 - -10°C for meteoric ice) (e.g., Rist et al., 2002). Temperatures of >-10°C are rather found in shallow, or the deepest parts of deep ice sheets, close to bedrock. There are studies on warm ice from glaciers (e.g., Hellmann TCD https://tc.copernicus.org/preprints/tc-2020-133/) so this shouldn't be mixed up. This is especially important since samples from set 2 did not match the desired outcome, thus there are still limits to the usability of this method. On p. 14 l.16 you state that the microstructure is "failing to match those conducted entirely at -10°C". In the conclusion on p. 14 l. 40 this is slightly emphasized by stating that the microstructure is "not [..] truly representative", please clarify this. Deformation mechanism maps might help to bring together different regimes (glaciers, ice sheets, ice shelves) e.g., RX diagram in Faria et al. (2014b), Frost & Ashby ( http://engineering.dartmouth.edu/defmech/) and Shoji and Higashi (1978, https://doi.org/10.3189/S002214300003358X). This might go beyond the scope of the manuscript, but should be kept in mind.

10. Tests were conducted on laboratory ice only. Natural ice has different, and highly variable, properties regarding e.g., absolute impurity content and spatial distribution of inclusions (cloudy bands) etc., which are reported to affect the rheological parameters, and thus the deformation, of ice. It should be emphasized in the discussion and the conclusions that the discussed results are not as simply transferable to natural ice as currently concluded (p.15 l.45). The study is an important step forward, but more research is needed to verify the easy upscaling to ice sheets and ice shelves.

11. 2.1 Laboratory: After cutting and polishing the samples, were they left for sublimation? Please address this issue briefly in the text since this can have an impact on the texture (grain shape and size) and on the quality of the FA measurements.

12. P. 12 Fig 5: LC023 has visibly, and measurably, much smaller grains than LC021, LC025, and LC026 and a rather homologous bulk grain size. This should be briefly

mentioned in the final section of the results indicating the small-scale differences in polycrystalline ice.

13. I suggest to combine the first sentences on p. 13 l. 47ff or to add some references in the first sentence.

14. References needed in final discussion paragraph on p. 14 l. 17fff discussing GBM, BLG, and other lower-temperature experiments.

15. Fig. 4 and Fig. 5: What is the reason for showing the c-axis orientation of 5000 pixels rather than using the c-axes of the actual grains as described for deriving the CPO in section 2.3? The number of grains is lower than 5000 and CPO contour plots thus probably look different when referring to the grains.

16. Appendix p. 15 l.5f. What is the reason to remove elongated small grains? Is it motivated by the possibility of artificially introduced grains due to segmentation/grain reconstruction? Were images manually checked for such grains? Please explain.

17. Do you have ideas what could have caused the two troughs in strain rate observed at LC025 at -10°C? (Fig.3d)

References: D. Dahl-Jensen, K. Mosegaard, N. Gundestrup, G. D. Clow, S. J. Johnsen, A. W. Hansen and N. Balling. Past Temperatures Directly from the Greenland Ice Sheet. Science 282 (5387), 268-271. DOI: 10.1126/science.282.5387.268

FARIA, Sérgio H.; WEIKUSAT, Ilka; AZUMA, Nobuhiko. The microstructure of polar ice. Part II: State of the art. Journal of structural geology, 2014, 61. Jg., S. 21-49.

Llorens, M. G., Griera, A., Bons, P. D., Roessiger, J., Lebensohn, R., Evans, L., & Weikusat, I. (2016). Dynamic recrystallisation of ice aggregates during co-axial visco-plastic deformation: a numerical approach. Journal of Glaciology, 62(232), 359-377.

Llorens, M. G., Griera, A., Bons, P. D., Lebensohn, R. A., Evans, L. A., Jansen, D., & Weikusat, I. (2016). Full-field predictions of ice dynamic recrystallisation under simple

shear conditions. Earth and Planetary science letters, 450, 233-242.

Mony L, Roberts JL, Halpin JA (2020). Inferring geothermal heat flux from an ice-borehole temperature profile at Law Dome, East Antarctica. Journal of Glaciology 66(257), 509–519. https://doi.org/10.1017/jog.2020.27

Rist, M. A., Sammonds, P. R., Oerter, H., & Doake, C. S. M. (2002). Fracture of Antarctic shelf ice. Journal of Geophysical Research: Solid Earth, 107(B1), ECV-2.

ShÅ■ji, H., & Higashi, A. (1978). A Deformation Mechanism Map of Ice. Journal of Glaciology, 21(85), 419-427. doi:10.3189/S002214300003358X

Technical corrections:

1. Author list: Jason Roberts4,6

2. wording sometimes not consistent e.g., behaviour and behavior, e.g. and e.g.,

3. Abbreviations sometimes inconsistent: CPO defined on p. 2 l. 29 and on p. 3 l. 52 GBM defined on p.4 l. 9 and on p. 13 l. 54. Shortly after introducing the abbreviation on p.4 l. 9 it is spelled out completely (p.4 l. 12)

4. Fig. 1 and Table 1: Units are placed after a comma, in the other plots units are encased by brackets

5. Suggestions for Fig. 2: grey temperature data could be a bit stronger, it's not easy to see in a printed version 2a: the x 10ˆ-8 might be placed somewhere else to increase visibility

6. P. 4 l. 64 one bracket too much

7. P.8 l.72: (LC001, LC002 and LC009)

8. P. 8 l.94 higher

9. Fig. 4 and 5: there is enough space to enhance the font size of the text in the legend in the upper right (text in Schmidt plot and parallel to the density colour scale).

(Contour plots) instead of (Conotur plots)

10. Appendix A: p.15 l. 51 G60 or G50 data? The Fabric Analyzer G60 has already been used, but you probably refer to the FA G50 here.
* * *

---

## Referee Comment (RC2) · Anonymous Referee #2 · 2 Jan 2021

Review of Craw et al. "The temperature change shortcut: effects of mid-experiment temperature changes on the deformation of polycrystalline ice"

The authors present results from a novel experiment aiming to determine if ice deformation experiments can be sped up by starting deformation tests at higher temperatures, when looking for tertiary strain rates (traditionally very tedious measurements to ascertain). These are exciting results, as they suggest that deformation test times can indeed be decreased by initiating experiments at higher temperatures for relatively high-temperature experiments. Below are some general and specific comments that may improve the paper.

[Figure]

General Comments: - This is a very well written paper that is clear and easy to follow. Additionally, the figures presented are also easy to interpret, visually pleasing, and appropriately support the text of the manuscript.

- In the Mechanical results section (Section 3.1), the authors state that the tertiary strain rates achieved in the constant-temperature and changing-temperature experiments within each Set are "within the level of variation between the duplicate experiments". From Figure 3 and Table 1, it appears that very similar tertiary strain rates were achieved in all of the -2C and -7C scenarios, but in the -10C scenario it appears that the changing-temperature experiment achieved a lower tertiary strain rate compared to the constant-temperature experiment. However, the authors state in the Discussion that the "Tertiary strain rates at both -7C and -10C from the changing-temperature experiments agree with those from their equivalent constant-temperature experiments to within the same level of variability..." Please give further justification for this result description and subsequent discussion of the -10C scenario.

Specific Comments: - How were the times and amount of added weight chosen to "periodically increase the loads"?

- Why were the samples kept in the rig setup after the experiments ended? Perhaps to not disturb the other experiments going on. Can you give some clarification on why lowering the temperatures to -18C and leaving the samples for days does not impact any final microstructure measurements.

- Why do the authors think that the resulting grain size (gs_med) is so different for the two -7C samples (LC004 and LC005), and the two -10C samples (LC021 and LC023)? Similarly, why did the two temperature-change experiment samples in Set 1 (LC006 and LC007) have such a large discrepancy in the resulting grain size? In the Discussion, the authors describe these sets of samples as having indistinguishable microstructure. Therefore, what range of grain size (gs_med) values is considered "indistinguishable", or similar enough? This information will be helpful to the reader

while interpreting Figures 4&5, and Table 1.

Section 1.2, Line 56: unclear what "… this can be delayed significantly beyond the establishment of a quasi-constant strain rate" means. Perhaps consider making this a separate sentence, such as "However, the formation of this steady-state microstructure can occur significantly after the establishment of a quasi-constant strain rate."

Section 1.3, Line 7: remove extra "and" from "… deformed ice and quartz aggregates and quartz veins…"

Section 1.4, Lines 29-33: awkward, long sentence. Consider rewording, especially the transition "…through to tertiary strain, if it is deformed…"

Section 1.4, Lines 44-45: change the sentence to use either the prepositions "with" or "to" after "compare" (… to compare X with/to Y…)

Section 2.1, Line 56: change "…frozen into…" to "frozen onto"; also, do you mean aluminum plates instead of "platens"?

Section 2.1, Lines 76-82: These are experimental results and should be moved to the Results section.

Section 3.2, Lines 34-35: It would be nice to see this result visually, instead of taking the authors' word for it.

Appendix A, Line 51: Do the authors mean "G50" here instead of "G60"?

Figure 1 caption: move "(a)" and "(b)" labels to before the panel descriptions.

Figure A1 caption: period at the end of the caption needed.

References: Hammonds & Baker (2018), Qi et al. (2019) not cited in the text.
* * *

---

## Author Comment (AC1) · 24 Feb 2021

We would like to thank Nicolas Stoll for his very helpful and constructive comments. We have responded to each comment in turn below, and describe changes we have made to the manuscript to address the issues raised.

[Figure]

**Reviewer 1**

1. **Abstract p1. L.2 We can do this by... Is "this" referring to understanding the mechanical properties of flowing ice, modelling of the dynamics of ice sheets, or predicting the behaviour in the future? Please rephrase to enhance clarity.**

   We have changed "We can do this..." to "We can increase our understanding of ice physical properties...".

2. **Abstract p.1 l.4 conditions in ice sheets and ice shelves extend to low temperatures (<-5 °C). The temperatures in the majority of ice sheets and ice shelves is well below -5 °C, a lower temperature value might be more appropriate. See more detailed comment below.**

   This is a good point. We have changed "<-5 °C" to "«-10 °C"

3. **Aims of the study are explained on p.2 l.32 and on p.5 l.43. They differ in details (e.g., tertiary creep only mentioned on p.2. l.32), thus it might be appropriate to combine both sections into one, placed at the end of the introduction to increase readability.**

   We removed the last sentence of section 1.1, and combined with the final sentence of the introduction as suggested.

4. **P.2 l.45 Primary creep: The definition is a little bit too simplified, maybe include some information from e.g., Faria et al. (2014b)...**

   We inserted more detail from (Faria, et al., 2014) as suggested: "strain rate decreases rapidly due to work hardening, as strain incompatibilities between grains and the load transfer from easy-glide to hard-glide systems result in heterogeneous internal stresses and the formation of dislocation tangles and subgrain

boundaries (Faria et al., 2014). The decreasing rate of deformation is controlled by crystals which are unfavourably oriented for creep (Duval et al., 1983)."

5. **P4. L10: The impact of GBM is strong on texture (grain size, grain shape, SPO) and on grain growth and reduction (dynamic grain growth, see Steinbach et al., 2017), but not so much on CPO (fabric)(e.g., Llorens et al., 2016a, b). Terminology is not always used in the same way in earth and material sciences, so please define "microstructure" in the beginning. Otherwise it is difficult to distinguish between microstructure (glaciology: fabric + texture), fabric and texture (see also comment 8).**

Good point. We are using the word "microstructure" in the glaciological sense, incorporating fabric and texture. We have added this definition in the text (now pg.2, L4-5).

6. **P. 5 l. 25: references for examples of other experiments, in-situ data, extrapolations missing.**

To incorporate referenced examples, we have changed this passage to read:

"Consequently, the majority of experimental ice deformation studies are performed at temperatures of $>-10\,^{\circ}\text{C}$, at a narrow range of stresses (e.g. Kamb, 1972; Jacka, 1984; Wilson et al., 2014; Montagnat et al., 2015), with a much smaller number of studies extending to lower temperatures and higher stresses (e.g. Goldsby and Kohlstedt, 2001; Wilson and Peternell, 2012; Qi et al., 2017). This means there is a bias in the available data favouring a small range of conditions which are seldom present in nature."

7. **P. 5 l. 43 it should be clarified that the systematic study is undertaken on laboratory ice.**

Changed to specify "laboratory ice".

8. **It is mentioned on p. 9 l. 101 that SPO data was derived, but this data is not further used in the manuscript. Showing and discussing SPO data would be a good way to further quantify the microstructural changes, i.e. changes in the ice texture. In this case it is possible to visually analyse the microstructure of the thin section images, however, I would suggest to add SPO data to fully fulfil the statement of "quantifiable differences in the microstructure" as described in the abstract on p.1 l. 14. Otherwise, if only CPO-data is used to quantify changes I suggest to rephrase the wording to e.g., "quantifiable differences in the fabric" to avoid confusion.**

Initially we had left the SPO data out as it does not show anything immediately relevant to the discussion, but you are right that it is good to include for completeness. We have added it as an appendix alongside grain size statistics, as suggested by reviewer 2, and mentioned it in the results section.

9. **For practical reasons the temperatures of the conducted experiments are rather high compared to temperatures in deep ice sheets (-30 °C- -20 °C) (e.g., Dahl-Jensen et al., 1998, Mony et al. (2020) and, to a lesser degree, in ice shelves (-25 - -10 °C for meteoric ice) (e.g., Rist et al., 2002). Temperatures of >-10 °C are rather found in shallow, or the deepest parts of deep ice sheets, close to bedrock. There are studies on warm ice from glaciers (e.g., Hellmann TCD https://tc.copernicus.org/preprints/tc-2020-133/) so this shouldn't be mixed up. This is especially important since samples from set 2 did not match the desired outcome, thus there are still limits to the usability of this method. On p. 14 l.16 you state that the microstructure is "failing to match those conducted entirely at -10 °C". In the conclusion on p. 14 l. 40 this is slightly emphasized by stating that the microstructure is "not [..] truly representative", please clarify this. Deformation mechanism maps might help to bring together different regimes (glaciers, ice sheets, ice shelves) e.g., RX diagram in Faria et al. (2014b), Frost and**

**Ashby (http://engineering.dartmouth.edu/defmech/) and Shoji and Higashi (1978, https://doi.org/10.3189/S002214300003358X). This might go beyond the scope of the manuscript, but should be kept in mind.**

You're right that it's important to make this disctinction more clearly, and remind the reader that at even lower temperatures the balance of deformation mechanisms will be different. We have added a sentence at the end of section 4: "The temperatures we have tested here are comparable to those found in temperate glaciers, and in the lower and upper extremities of polar ice sheets. For experiments aiming to replicate colder conditions, it would be best to use a lower starting temperature, so that the balance of deformation mechanisms active at the beginning of the experiment is more comparable to that at the final target temperature.".

10. **Tests were conducted on laboratory ice only. Natural ice has different, and highly variable, properties regarding e.g., absolute impurity content and spatial distribution of inclusions (cloudy bands) etc., which are reported to affect the rheological parameters, and thus the deformation, of ice. It should be emphasized in the discussion and the conclusions that the discussed results are not as simply transferable to natural ice as currently concluded (p.15 l.45). The study is an important step forward, but more research is needed to verify the easy upscaling to ice sheets and ice shelves.**

We have added a sentence in the discussion: "It should also be noted that natural ice has different rheological properties to standard ice, (Budd and Jacka, 1989; Dahl-Jensen et al., 1997; Castelnau et al., 1998; Craw et al., 2018), and so the balance of deformation mechanisms active in experiments may be different to those active under the same conditions in nature."

We have also changed the concluding remarks to separate the specific contribution of this study (extending the temperature conditions that are feasible for laboratory experiments by reducing experiment time) and the hopeful outcome of

this (allowing experiments to become more representative of *in situ* conditions, therefore leading to more accurate flow law parameters).

11. **2.1 Laboratory: After cutting and polishing the samples, were they left for sublimation? Please address this issue briefly in the text since this can have an impact on the texture (grain shape and size) and on the quality of the FA measurements.**

Samples were cut into thin sections with a microtome, and then repeatedly thinned and checked under cross-polarised light until birefringence was minimal. Occasionally where the fabric analyser data was of lower quality, then section was left for ~1hr to sublimate and then the scan was repeated. We have clarified this in the methods.

12. **P. 12 Fig 5: LC023 has visibly, and measurably, much smaller grains than LC021, LC025, and LC026 and a rather homologous bulk grain size. This should be briefly mentioned in the final section of the results indicating the small-scale differences in polycrystalline ice.**

Yes, this is a good example of the possible variability between experiments conducted under the same conditions. See our more in-depth response to Reviewer 2, comment #4, below.

13. **I suggest to combine the first sentences on p. 13 l. 47 or to add some references in the first sentence.**

We have changed this to: "The microstructural characteristics observed in these samples after deformation are comparable to those from other compression experiments in the literature; the development of a vertical small-circle girdle CPO centred around the compression direction has been observed many times in polycrystalline ice above $-15\,°C$ (e.g. Kamb, 1972; Jacka, 1984; Treverrow et al.,2012; Wilson et al., 2014), and the interlocking, irregular grain boundaries

seen in all deformed samples in this study are comparable to those observed by Montagnat et al. (2015) and Jacka and Jun (1994) after similar experiments."

14. **References needed in final discussion paragraph on p. 14 l. 17 discussing GBM, BLG, and other lower-temperature experiments.**

Added references to Alley, 1992; Montagnat et al., 2015; Qi et al., 2017.

15. **Fig. 4 and Fig. 5: What is the reason for showing the c-axis orientation of 5000 pixels rather than using the c-axes of the actual grains as described for deriving the CPO in section 2.3? The number of grains is lower than 5000 and CPO contour plots thus probably look different when referring to the grains.**

Well spotted, the statement in section 2.3 is incorrect.. We ultimately chose to use randomised pixel data for the CPO because, while we do remove all very fine and low geometric quality grains, smaller grains have a higher chance of being artefacts of the data processing. Plotting one point per grain would amplify the contribution of smaller grains to the CPO data. In fact the difference is very minimal, and so for our purposes it is a fairly arbitrary choice (see figure 1).

We have corrected this in section 2.3.

16. **Appendix p. 15 l.5f. What is the reason to remove elongated small grains? Is it motivated by the possibility of artificially introduced grains due to segmentation/grain reconstruction? Were images manually checked for such grains? Please explain.**

Thank you for pointing this out. The elongated small grains are a common artefact in two-dimensional ice deformation experiments, which we were working on while we developed the process, and they are not present here. We have removed that step, and the results are unchanged.
17. **Do you have ideas what could have caused the two troughs in strain rate observed at LC025 at -10 °C? (Fig.3d)**

All of the raw strain rate data have these periodic jumps, which appear when we increase the loads to approximate constant stress (see section 2.1). For the most part they are obvious as an instantaneous increase in displacement, and are straightforward to remove during data processing to avoid large artefacts in the results. However, they are often accompanied by more complicated disturbances to the apparatus (adjusting other equipment, opening and closing neighbouring freezers, etc). This means that some of these jumps are harder to remove without "doctoring" the data, and so we err on the side of caution when trying to correct them. Another clear example is in LC009 at a strain of $\sim 0.07$. We have clarified this slightly in section 2.2: "Sudden jumps in displacement from load increases and disturbances to the apparatus were removed manually. Some more gradual jumps in displacement (e.g. in **LC021** and **LC025**) could not be easily removed, and so were left to avoid overprocessing the data."

- **Technical corrections**

  All corrected, thank you.

---

## Author Comment (AC2) · 24 Feb 2021

We would like to thank the anonymous reviewer for their insightful comments and questions. We have responded to each comment in turn below, and describe changes we have made to the manuscript to address the issues raised.

[Figure]

**Reviewer 2**

1. **In the Mechanical results section (Section 3.1), the authors state that the tertiary strain rates achieved in the constant-temperature and changing-temperature experiments within each Set are "within the level of variation between the duplicate experiments". From Figure 3 and Table 1, it appears that very similar tertiary strain rates were achieved in all of the -2C and -7C scenarios, but in the -10C scenario it appears that the changing-temperature experiment achieved a lower tertiary strain rate compared to the constant-temperature experiment. However, the authors state in the Discussion that the "Tertiary strain rates at both -7C and -10C from the changing-temperature experiments agree with those from their equivalent constant-temperature experiments to within the same level of variability..." Please give further justification for this result description and subsequent discussion of the -10C scenario.**

This is a good point which needed clarifying in the text. You are right that the changing-temperature experiments at -10 °C have a lower tertiary strain rate, and we believe this is due to a limitation with the experiments rather than a real effect. The experiments which ran to strains greater than $\sim 0.08$ show a steady drop-off in strain rate after that point, which we believe is because our assumption that cross-sectional area is increasing consistently down the length of the sample becomes less valid (see our response to 2, below). Because the constant-temperature $-10\,^\circ$C experiments (LC021 and LC023) took such a long time, we were only able to run them to the very beginning of tertiary creep, while the other experiments could run to higher strains. As a result, they never experienced the drop-off in strain rates, while their changing-temperature counterpart experiments reached higher strains and so did begin to decrease in strain rate. As we describe in section 2.2, representative tertiary strain rate values for the experiments were taken from earlier in the experiment, at the beginning of tertiary creep, where the

data are more robust. In figure 1 (attached), we have projected the curves for LC021 and LC023 forward with the same curvature as would be expected based on results from the other experiments. As highlighted by the shaded rectangles, the difference in final tertiary strain rates between the changing-temperature and constant-temperature experiments at both -7 °C and -10 °C is very similar. In both cases in fact strain rates are lower in the changing-temperature experiments, but this difference is small and does not exceed the level of variability expected in these kinds of experiments ($\pm 20\%$), so we are not able to attribute it to any real mechanism.

We have clarified this in the results section of the manuscript (now pg. 9, L26-29).

2. **How were the times and amount of added weight chosen to "periodically increase the loads"?**

We aim to approximate a constant compressive stress, using weights loaded on top of the rigs. As the samples are compressed, they increase in cross-sectional area, and so we add an amount of weight which will increase the stress back to the target stress, assuming that volume is conserved and that the sample is expanding horizontally at the same rate everywhere (this assumption becomes less valid as strain increases, as the middle of the sample is in fact expanding at a higher rate than the top and bottom). Loads were increased every 2-5 days depending on the strain rate (higher strain rates mean it is a shorter time before the increase in cross-sectional area is significant). This is frequently enough to minimise drops in stress due to sample expansion, but infrequently enough to minimise disturbances to the experiments. We have clarified this slightly in the methods (now pg. 7, L14).

3. **Why were the samples kept in the rig setup after the experiments ended? Perhaps to not disturb the other experiments going on. Can you give some clarification on why lowering the temperatures to -18C and leaving the sam-**

**ples for days does not impact any final microstructure measurements.**

Extracting the samples from the rigs involves multiple people, as the rigs must be lifted fully out of the freezers and brought quickly into a cold room for the sample to be detached and cut. It was easiest to wait for more than one experiment to finish and extract those samples at the same time, so some were left after the experiment was over until it was convenient to remove them. They were left with the weights still on to avoid any relaxation affecting the microstructure, only the temperature was changed. Essentially this temperature drop simply slowed the experiment down to the point that any further increase in strain was insignificant.

To give a rough idea, if we interpolate between the strain rate data for different applied shear stresses at -15 °C and -20 °C in Budd and Jacka (1989), with a shear stress of 0.25MPa at -18 °C we would expect to see a strain rate on the order of 2.5e-09s$^{-1}$. Over a full seven days (the maximum time we would leave an experiment before extraction), that would result in $\sim 0.0015$ of additional accumulated strain, which is insignificant compared with the $\sim 0.1$ already accumulated. We have added a brief explanation into the methods section (now pg. 8, L3-4).

4. **Why do the authors think that the resulting grain size (gs_med) is so different for the two -7C samples (LC004 and LC005), and the two -10C samples (LC021 and LC023)? Similarly, why did the two temperature-change experiment samples in Set 1 (LC006 and LC007) have such a large discrepancy in the resulting grain size? In the Discussion, the authors describe these sets of samples as having indistinguishable microstructure. Therefore, what range of grain size (gs_med) values is considered "indistinguishable", or similar enough? This information will be helpful to the reader while interpreting Figures 4 and 5, and Table 1.**

This important point was also raised by Reviewer 1, comment #12. You are right that there is a significant difference in grain sizes between samples deformed under the same conditions. This is most likely a sampling effect, as our thin

sections are small in area and the width of the samples prohibits the taking of multiple sections, so the grain size indicated by one one-dimensional section is not necessarily representative of the entire sample. Whilst the grain sizes seen in the $-10\,°\mathrm{C}$ samples do appear to be tending smaller than the higher temperature samples, the ranges of grain sizes seen in each temperature set overlap with one another, and so it is not possible to actually distinguish the sample groups based on grain size data from such a small number of experiments. This is what we mean by "...it is not possible to distinguish between changing-temperature and constant-temperature experiments in Set 1 on the basis of microstructure".

We have clarified this at the end of the results section: "Because thin section measurements of grain size sample only a small number of grains, and there is a large range of grain sizes measured in samples deformed under the same conditions, we are unable to draw any conclusions based on grain size differences between sets."

5. **Section 1.2, Line 56: unclear what "...this can be delayed significantly beyond the establishment of a quasi-constant strain rate" means. Perhaps consider making this a separate sentence, such as "However, the formation of this steady-state microstructure can occur significantly after the establishment of a quasi-constant strain rate."**

Good suggestion. We have changed this to: "However, the formation of this steady-state microstructure can occur much later, after the establishment of a quasi-constant strain rate. While tertiary creep..."

6. **Section 1.4, Lines 29-33: awkward, long sentence. Consider rewording, especially the transition "...through to tertiary strain, if it is deformed..."**

This unwieldy sentence has been replaced with two:

"Studies in both natural ice (Russell-Head and Budd, 1979; Gao and Jacka, 1987) and laboratory ice (Treverrow et al., 2012) have deformed samples through to

tertiary strain, and then deformed them again at a later stage under the same conditions. In these cases, the second deformation phase of the experiments progresses straight from the initial elastic deformation stage to resume deformation at the same constant tertiary strain rate, with no significant change in CPO, allowing tertiary creep to be reached within strains of $2 - 3\%$. However, if the stress configuration is changed in the second stage of the experiment, characteristics of the original CPO can persist to higher strains (Budd and Jacka, 1989)"

7. **Section 1.4, Lines 44-45: change the sentence to use either the prepositions "with" or "to" after "compare" (...to compare X with/to Y...)**

Changed to "to" as suggested.

8. **Section 2.1, Line 56: change "...frozen into..." to "frozen onto"; also, do you mean aluminum plates instead of "platens"?**

"Platen' is a word for a mounting plate for materials being pressed or deformed, more commonly used in engineering and manufacturing. It's not in common usage, so we've replaced it with "mount" to avoid confusion. The platen/mount contains a large depression which the ice is actually frozen into. We have clarified this.

9. **Section 3.2, Lines 34-35: It would be nice to see this result visually, instead of taking the authors' word for it.**

We have added these data in Appendix B.

10. **Appendix A, Line 51: Do the authors mean "G50" here instead of "G60"?**

Our meaning was that FAME can be used with G60 data, but the method described here can be used with G50 data. This was poorly worded, so we've removed it.

11. **Figure 1 caption: move "(a)" and "(b)" labels to before the panel descriptions.**

    Changed as suggested.

12. **Figure A1 caption: period at the end of the caption needed.**

    Changed as suggested.